# R2D2: Repeatable and Reliable Detector and Descriptor

**Jerome Revaud**     **Philippe Weinzaepfel**     **César De Souza**     **Martin Humenberger**

NAVER LABS Europe

`firstname.lastname@naverlabs.com`

## Abstract

Interest point detection and local feature description are fundamental steps in many computer vision applications. Classical approaches are based on a *detect-then-describe* paradigm where separate handcrafted methods are used to first identify repeatable keypoints and then represent them with a local descriptor. Neural networks trained with metric learning losses have recently caught up with these techniques, focusing on learning repeatable saliency maps for keypoint detection or learning descriptors at the detected keypoint locations. In this work, we argue that repeatable regions are not necessarily discriminative and can therefore lead to select suboptimal keypoints. Furthermore, we claim that descriptors should be learned only in regions for which matching can be performed with high confidence. We thus propose to jointly learn keypoint detection and description together with a predictor of the local descriptor discriminativeness. This allows to avoid ambiguous areas, thus leading to reliable keypoint detection and description. Our *detection-and-description* approach simultaneously outputs sparse, repeatable and reliable keypoints that outperforms state-of-the-art detectors and descriptors on the HPatches dataset and on the recent Aachen Day-Night localization benchmark.

## 1 Introduction

Accurately finding and describing similar points of interest (*keypoints*) across images is crucial in many applications such as large-scale visual localization [46, 56], object detection [7], pose estimation [32], Structure-from-Motion (SfM) [50] and 3D reconstruction [22]. In these applications, extracted keypoints should be sparse, repeatable and discriminative in order to maximize the matching accuracy with a low memory footprint.

Classical approaches are based on a two-stage pipeline that first detects keypoints [18, 27, 28, 29] and then computes a local descriptor for each keypoint [4, 25]. Specifically, the role of the keypoint detector is to find scale-space locations with covariance with respect to camera viewpoint changes and invariance with respect to photometric transformations. A large number of handcrafted keypoints have shown to work well in practice, such as corners [18] or blobs [25, 27, 28]. As for the description, various schemes based on histograms of local gradients [4, 6, 24, 43], whose most well known instance is SIFT [25], were proposed and are still widely used.

Despite this apparent success, this paradigm was recently challenged by several data-driven approaches willing to replace the handcrafted parts [14, 17, 26, 30, 33, 35, 49, 58, 59, 60, 63, 65]. Arguably, handcrafted methods are limited by the *a priori* knowledge researchers have about the tasks at hand. The point is thus to let a deep network automatically discover which feature extraction process and representation are most suited to the data. The few attempts for learning keypoint detectors [9, 11, 14, 35, 49, 63] have only focused on the repeatability. On the other hand, metric learning techniques applied to learning local robust descriptors [26, 33, 58, 59] have recently outperformed traditional descriptors, including SIFT [21]. They are trained on the repeatable locations provided by the detector, which may harm the performance in regions that are repeatable but where accurate

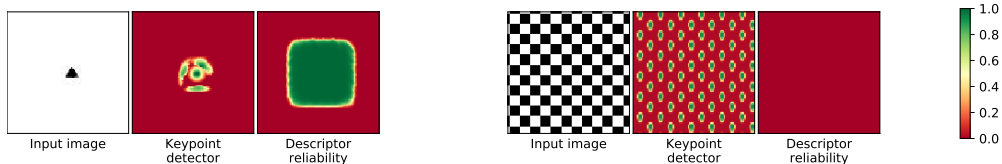

Figure 1: Toy examples to illustrate the key difference between repeatability (2nd column) and reliability (3rd column) for a given image. Repeatable regions in the first image are only located near the black triangle, however, all patches containing it are equally reliable. In contrast, all squares in the checkerboard pattern are salient hence repeatable, but are not discriminative due to self-similarity.

matching is not possible. Figure 1 shows such an example with a checkerboard image: every corner or blob is repeatable but matching cannot be performed due to the repetitiveness of the pattern. In natural images, common textures such as the tree leafage, skyscraper windows or sea waves can be salient but hard to match because of their repetitiveness and unstable nature.

In this work, we claim that detection and description are inseparably tangled since good keypoints should not only be *repeatable* but should also be *reliable* for matching. We thus propose to jointly learn the descriptor reliability seamlessly with the detection and description processes. Our method estimates a confidence map for each of these two aspects and selects only keypoints which are both repeatable and reliable. More precisely, our network outputs dense local descriptors (one for each pixel) as well as two associated repeatability and reliability confidence maps. The two maps respectively aim to predict if a keypoint is repeatable and if its descriptor is discriminative, *i.e.*, if it can be accurately matched with high confidence. Our keypoints thus correspond to locations that maximize both confidence maps.

To train the keypoint detector, we employ a novel unsupervised loss that encourages repeatability, sparsity and a uniform coverage of the image. As for the local descriptor, we introduce a new loss to learn reliable local descriptors while specifically targeting image regions that are meaningful for matching. It is trained with a listwise ranking loss based on a differentiable Average Precision (AP) metric, hereby leveraging recent advances in metric learning [5, 21, 39]. We jointly learn an estimator of the descriptor reliability to predict which patches can be matched with a high AP, *i.e.*, that are both discriminative, robust and in the end that can be accurately matched. Experiment results show that our elegant formulation of joint detector and descriptor selects keypoints which are both repeatable and reliable, leading to state-of-the-art results on the HPatches and Aachen datasets. Our code and models are available at `https://github.com/naver/r2d2`.

## 2   Related work

Local feature extraction and description have received a continuous influx of attention in the past several years (*cf.* surveys in [8, 13, 44, 61]). We focus here on the learning methods only.

**Learned descriptors.**   Most deep feature matching methods have focused on learning the descriptor component, applied either on a sparse set of keypoints [3, 26, 30, 54, 55] detected using standard handcrafted methods or densely over the image [12, 33, 48, 57]. The descriptor is usually trained using a metric learning loss that seeks to maximize the similarity of descriptors corresponding to the same patches and minimize it otherwise [1, 17, 26, 58, 59]. To this aim, the triplet loss [15, 53] and the contrastive loss [38] have been widely used: they process two or three patches at a time, steadily optimizing the global objective based on local comparisons. Another type of loss, labeled as *global* in opposition, have been recently proposed by He *et al.* [21]. Inspired by advances in listwise losses [20, 62], it consists in a differentiable approximation of the Average-Precision (AP), a standard ranking metric evaluating the global ranking, which is directly optimized during training. It was shown to produce state-of-the-art results in patch and image matching [5, 21, 39]. Our approach also optimizes the AP but has several advantages over [21]: (a) the detector is trained jointly with the descriptor, alleviating the drawbacks of sparse handcrafted keypoint detector; (b) our approach is fully convolutional, outputting dense patch descriptors for an input image instead of being applied patch by patch; (c) our novel AP-based loss jointly learns patch descriptors and an estimate of their reliability, allowing in turn the network to minimize its effort on undistinctive regions.

**Learned detectors.** The first approach to rely on machine learning for keypoint detection was FAST [42]. Later, Di *et al.* [10] learn to mimic the output of handcrafted detectors with a compact neural network. In [23], handcrafted and learned filters are combined to detect repeatable keypoints. These two approaches still rely on some handcrafted detectors or filters while ours is trained end-to-end. QuadNet [49] is an unsupervised approach based on the idea that the ranking of the keypoint salience are preserved by natural image transformations. In the same spirit, [64] additionally encourage peakiness of the saliency map for keypoint detector on textures. In this paper, we employ a simpler unsupervised formulation that locally enforces the similarity of the saliency maps.

**Jointly learned descriptor and detector.** In the seminal LIFT approach, Yi *et al.* [63] introduced a pipeline where keypoints are detected and cropped regions are then fed to a second network to estimate the orientation before going throughout a third network to perform description. Recently, the SuperPoint approach by DeTone *et al.* [9] tackles keypoint detection as a supervised task learned from artificially generated training images containing basic structures like corners and edges. After learning the keypoint detector, a deep descriptor is trained using a second network branch, sharing most of the computation. In contrast, our approach learns both of them jointly from scratch and without introducing any artificial bias in the keypoint detector, which is also achieved by Georgakis *et al.* [14] for the specific task of 3D matching from depth images by leveraging a region-proposal network. Using a large-scale dataset of annotated landmark images, Noh *et al.* [33] trained DELF, an approach targeted for image retrieval that learns local features as a by-product of a classification loss coupled with an attention mechanism. In comparison, our approach is unsupervised and trained with relatively little data. More similar to our approach, Mishkin *et al.* [31] recently leverage deep learning to jointly enhance an affine regions detector and local descriptors. Nevertheless, their approach is rooted on a handcrafted keypoint detector that generates seeds for the affine regions, thus not truly learning keypoint detection. More recently, D2-Net [11] uses a single CNN for joint detection and description that share all weights; the detection being based on local maxima across the channels and the spatial dimensions of the feature maps. Similarly, Ono *et al.* [35] train a network from pairs of matching images with a complicated asymmetric gradient backpropagation scheme for the detection and a triplet loss for the local descriptor.

Compared to these works, we highlight for the first time the importance of treating repeatability and reliability as separate entities represented by their own respective score maps. Our novel AP-based reliability loss allows us to estimate patch reliability according to the AP metric while simultaneously optimizing for the descriptor. In a single batch, each patch is typically compared to thousands of other patches. In contrast to Hartmann *et al.* [19] that predicts reliability given fixed descriptors, our novel loss tightly couples descriptors and reliability estimates. This capability cannot be achieved with the standard contrastive and triplet losses used in prior work. Overall, being able to train a keypoint detector from scratch while jointly predicting reliable descriptors is made possible by our novel losses that are unlike any of the ones used in [9, 11, 14, 21, 35, 49].

## 3 Joint learning reliable and repeatable detectors and descriptors

The proposed approach, referred to as R2D2, aims to predict a set of sparse locations of an input image $I$ that are repeatable and reliable for the purpose of local feature matching. In contrast to classical approaches, we make an explicit distinction between repeatability and reliability. As shown in Figure 1, they are in fact two complementary aspects that must be predicted separately.

We thus propose to train a fully-convolutional network (FCN) that predicts 3 outputs for an image $I$ of size $H \times W$. The first one is a 3D tensor $\boldsymbol{X} \in \mathbb{R}^{H \times W \times D}$ that corresponds to a set of dense D-dimensional descriptors, one per pixel. The second one is a heatmap $\boldsymbol{S} \in [0, 1]^{H \times W}$ whose goal is to provide sparse yet repeatable keypoint locations. To achieve sparsity, we only extract keypoints at locations corresponding to local maxima in $\boldsymbol{S}$. The third output is an associated reliability map $\boldsymbol{R} \in [0, 1]^{H \times W}$ that indicates the estimated reliability of descriptor $\boldsymbol{X}_{ij}$, *i.e.*, likelihood that it is good for matching, at each pixel $(i, j)$ with $i \in \{1, \ldots, W\}$ and $j \in \{1, \ldots, H\}$.

The network architecture is shown in Figure 2. The backbone is a L2-Net [58], with two minor differences: (a) subsampling is replaced by dilated convolutions in order to preserve the input resolution at all stages, and (b) the last $8 \times 8$ convolutional layer is replaced by 3 successive $2 \times 2$ convolutional layers. We found that this latter modification reduces the number of weights by a factor 5 for a similar accuracy. The 128-dimensional output tensor serves as input to: (a) a $\ell_2$-normalization

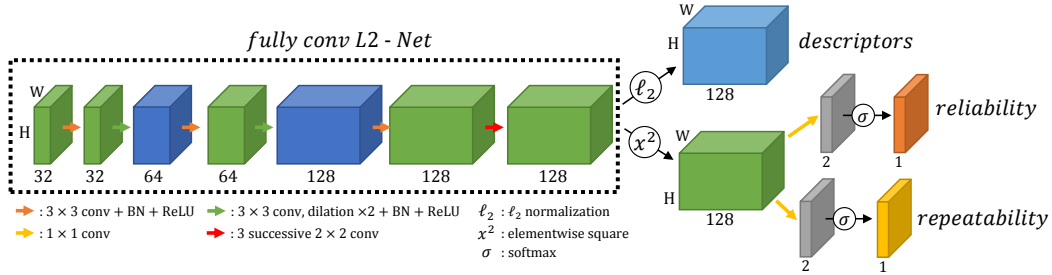

Figure 2: Overview of our network for jointly learning repeatable and reliable matches.

layer to obtain the per-pixel patch descriptors $\boldsymbol{X}$, (b) an element-wise square operation followed by an additional $1 \times 1$ convolutional layer and a softmax to obtain the repeatability map $\boldsymbol{S}$, and (c) an identical second branch to obtain the reliability map $\boldsymbol{R}$.

## 3.1 Learning repeatability

As observed in previous works [9, 63], keypoint repeatability is a problem that cannot be tackled by standard supervised training. In fact, using supervision essentially boils down in this case to imitating an existing detector rather than discovering potentially better keypoints. We thus treat the repeatability as a self-supervised task and train the network such that the positions of local maxima in $\boldsymbol{S}$ are covariant to natural image transformations like viewpoint or illumination changes.

Let $I$ and $I'$ be two images of the same scene and let $U \in \mathbb{R}^{H \times W \times 2}$ be the ground-truth correspondences between them. In other words, if the pixel $(i, j)$ in the first image $I$ corresponds to pixel $(i', j')$ in $I'$, then $U_{ij} = (i', j')$. In practice, $U$ can be estimated using existing optical flow or stereo matching if $I$ and $I'$ are natural images or can be obtained exactly if $I'$ was synthetically generated with a known transformation, *e.g.* an homography [9], see Section 3.3. Let $\boldsymbol{S}$ and $\boldsymbol{S}'$ be the repeatability maps for image $I$ and $I'$ respectively, and $\boldsymbol{S}'_U$ be $\boldsymbol{S}'$ warped according to $U$.

Ultimately, we want to enforce the fact that all local maxima in $\boldsymbol{S}$ correspond to the ones in $\boldsymbol{S}'_U$. Our key idea is to maximize the cosine similarity, denoted as $cosim$ in the following, between $\boldsymbol{S}$ and $\boldsymbol{S}'_U$. When $cosim(\boldsymbol{S}, \boldsymbol{S}'_U)$ is maximized, the two heatmaps are indeed identical and their maxima correspond exactly. While this is true in ideal conditions, in practice, local occlusions, warp artifacts or border effects make this approach unrealistic. Therefore we reformulate this idea *locally*, *i.e.*, we average the cosine similarity over many small patches. We define the set of overlapping patches $\mathcal{P} = \{p\}$ that contains all $N \times N$ patches in $\{1, \ldots, W\} \times \{1, \ldots, H\}$ and define the loss as:

$$\mathcal{L}_{cosim}(I, I', U) = 1 - \frac{1}{|\mathcal{P}|} \sum_{p \in \mathcal{P}} cosim\left(\boldsymbol{S}\left[p\right], \boldsymbol{S}'_U\left[p\right]\right), \tag{1}$$

where $\boldsymbol{S}\left[p\right] \in \mathbb{R}^{N^2}$ denotes the flattened $N \times N$ patch $p$ extracted from $\boldsymbol{S}$, and likewise for $\boldsymbol{S}'_U\left[p\right]$. Note that $\mathcal{L}_{cosim}$ can be minimized trivially by having $\boldsymbol{S}$ and $\boldsymbol{S}'_U$ constant. To avoid this, we employ a second loss function that aims to maximize the local peakiness of the repeatability map:

$$\mathcal{L}_{peaky}(I) = 1 - \frac{1}{|\mathcal{P}|} \sum_{p \in \mathcal{P}} \left( \max_{(i,j) \in p} \boldsymbol{S}_{ij} - \underset{(i,j) \in p}{\mathrm{mean}} \boldsymbol{S}_{ij} \right). \tag{2}$$

Interestingly, this allows to choose the spatial frequency of local maxima by varying the patch size $N$, see Section 4.2. Finally, the resulting repeatability loss is composed as a weighted sum of the first loss and second loss applied to both images:

$$\mathcal{L}_{rep}(I, I', U) = \mathcal{L}_{cosim}(I, I', U) + \frac{1}{2} \left( \mathcal{L}_{peaky}(I) + \mathcal{L}_{peaky}(I') \right). \tag{3}$$

## 3.2 Learning reliability

In addition to the repeatibility map $\boldsymbol{S}$, our network also computes dense local descriptors as well as a heatmap $\boldsymbol{R}$ that predicts the individual reliability $\boldsymbol{R}_{ij}$ of each descriptor $\boldsymbol{X}_{ij}$. The goal is to let the network learn to choose between making descriptors as discriminative as possible or, conversely, sparing its efforts on uniformative regions like the sky or the ground. To that aim, we propose a loss that is minimized when the network can successfully predict the actual descriptor *reliability*.

As in previous works [1, 17, 26, 58, 59], we cast descriptor matching as a metric learning problem. More specifically, each pixel $(i, j)$ from the first image $I$ is the center of a $M \times M$ patch $p_{ij}$ with descriptor $\boldsymbol{X}_{ij}$ that we can compare to the descriptors $\{\boldsymbol{X}'_{uv}\}$ of all other patches in the second image $I'$. Knowing the ground-truth correspondence mapping $U$, we estimate the reliability of patch $p_{ij}$ using the Average-Precision (AP), a standard ranking metric. We ideally want that patch descriptors are as reliable as they can be, *i.e.*, we want to maximize the AP for all patches. We therefore follow He *et al.* [21] and optimize a differentiable approximation of the AP, denoted as $\widetilde{\text{AP}}$. Training then consists in maximizing the AP computed for each of the $B$ patches $\{p_{ij}\}$ in the batch:

$$\mathcal{L}_{AP} = \frac{1}{B} \sum_{ij} 1 - \widetilde{\text{AP}}(p_{ij}). \tag{4}$$

Local descriptors are extracted at each pixel, but not all locations are equally interesting. In particular, uniform regions or elongated 1D patterns are known to lack the distinctiveness necessary for accurate matching [16]. More interestingly, even well-textured regions are also known to be unreliable from their unstable nature, such as tree leafages or ocean waves. It becomes thus clear that optimizing the patch descriptor even in such image regions can hinder performance. We therefore propose to enhance the AP loss to spare the network in wasting its efforts on undistinctive regions:

$$\mathcal{L}_{AP,\boldsymbol{R}} = \frac{1}{B} \sum_{ij} 1 - \widetilde{\text{AP}}(p_{ij})\boldsymbol{R}_{ij} + \kappa(1 - \boldsymbol{R}_{ij}), \tag{5}$$

where $\kappa \in [0, 1]$ is a hyperparameter that represents the AP threshold above which a patch is considered reliable. We found that $\kappa = 0.5$ yields good results in practice and we use this value in the rest of the paper. Figure 3.2 shows the loss function $\mathcal{L}_{AP,\boldsymbol{R}}$ for a given patch $p_{ij}$ as a function of $\widetilde{\text{AP}}(p_{ij})$ and $\boldsymbol{R}_{ij}$. For reliable patches (*i.e.* $AP > \kappa$), the loss incites to maximize the AP. Conversely, when $AP < \kappa$, the loss encourages the reliability to be low. This way, learning converges to a region where there is almost no gradients (at $\boldsymbol{R}_{ij} \simeq 0$), hence having barely any effect on descriptors that belong to undistinctive image regions. Note that

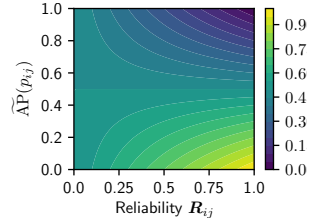

Figure 3: Visualization of our proposed loss $\mathcal{L}_{AP,\boldsymbol{R}}$.

a similar idea of jointly training the descriptor and an associated confidence was recently proposed in [34], but using a triplet loss, which prevents the use of an interpretable threshold $\kappa$ as in our case.

### 3.3 Inference and training details

**Runtime.** At test time, we run the trained network multiple times on the input image at different scales starting at $L = 1024$ pixels and downsampling it by $2^{1/4}$ each time until $L < 256$ pixels, where $L$ denotes the largest dimension of the image. For each scale, we find local maxima in $\boldsymbol{S}$ and gather descriptors from $\boldsymbol{X}$ at corresponding locations. Finally, we keep a shortlist of the best $K$ descriptors over all scales where the score of descriptor $\boldsymbol{X}_{ij}$ is computed as $\boldsymbol{S}_{ij}\boldsymbol{R}_{ij}$, *i.e.* requiring both repeatable and reliable keypoints. In practice, processing a 1M pixel image on a Tesla P100-SXM2 GPU takes about 0.5s to extract keypoints at a single scale (full image) and 1s for all scales.

**Training data.** We use three sources of data to train our method: (a) distractors from a retrieval dataset [37] (*i.e.*, random web images), from which we build synthetic image pairs by applying random transformations (homography and color jittering), (b) images from the Aachen dataset [45, 47], using the same strategy to build synthetic pairs, and (c) pairs of nearby views from the Aachen dataset where we obtain a pseudo ground-truth using optical flow (see below). All sources are represented approximately equally (about 4000 images each) and we study their importance in Section 4.4. Note that we *do not* use any image from the HPatches evaluation dataset [2] during training.

**Ground-truth correspondences.** To generate dense ground-truth correspondences between two images of the same scene, we leverage existing matching techniques. As in previous works [11, 35], we use points verified by Structure-from-Motion that we enhance by designing a pipeline based on optical flow tools to reliably extract dense correspondences. As a first step, we run a SfM pipeline [50] that outputs a list of 3D points and a 6D camera pose for each image. For each image pair with sufficient overlap (*i.e.*, with some common 3D points), we then compute the fundamental matrix. Next, we compute high-quality dense correspondences using EpicFlow [40]. We enhance it by adding epipolar constraints in DeepMatching [41], the first step of EpicFlow that produces semi-sparse

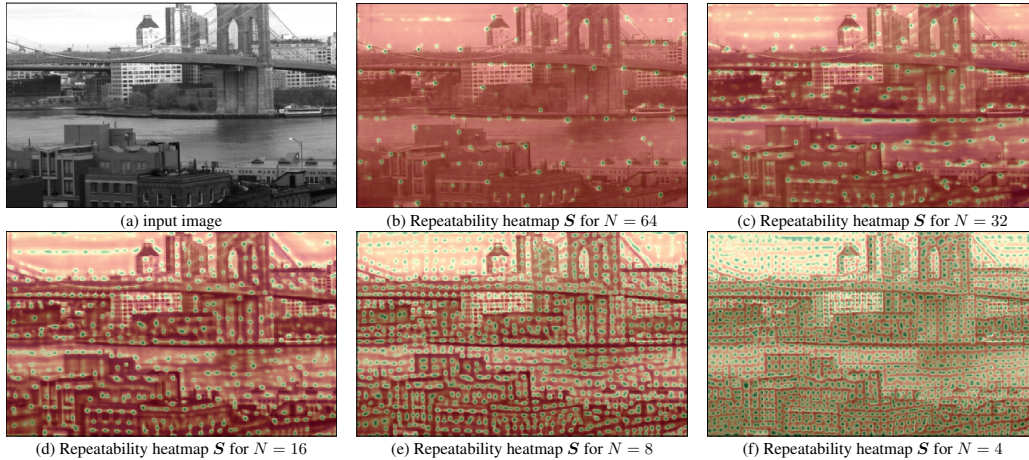

| (a) input image | (b) Repeatability heatmap $S$ for $N = 64$ | (c) Repeatability heatmap $S$ for $N = 32$ |
| (d) Repeatability heatmap $S$ for $N = 16$ | (e) Repeatability heatmap $S$ for $N = 8$ | (f) Repeatability heatmap $S$ for $N = 4$ |

Figure 4: Sample repeatability heatmaps obtained when training the repeatability loss $\mathcal{L}_{rep}$ from Eq. (3) with different patch size $N$. Red and green colors denote low and high values, respectively.

matches. In addition, we also predict a mask where the flow is reliable, as optical flow is defined at every pixel, even in occluded areas. We post-process the output of DeepMatching by computing a graph of connected consistent neighbors, and keeping only matches belonging to large connected components (at least 50 matches). The mask is defined using a thresholded kernel density estimator on the verified matches.

**Training parameters.** We optimize the network using Adam for 25 epochs with a fixed learning rate of 0.0001, weight decay of 0.0005 and a batch size of 8 pairs of images cropped to $192 \times 192$.

**Sampling issues for AP loss.** To have a setup as realistic as possible given hardware constraints, we subsample "query" patches in the first image on a regular grid of $8 \times 8$ pixels. To handle the inherent imperfection of the optical flow, we define a single positive per query patch $p_{ij}$ in the second image as the one with the most similar descriptor within a radius of 3 pixels from the ground-truth position $U_{ij}$. Negatives are defined as more than 5 pixels away from $U_{ij}$ and sampled on a $8 \times 8$ regular grid.

# 4 Experiments

## 4.1 Datasets and metrics

We evaluate our method on the full image sequences of the HPatches dataset [2]. The HPatches dataset contains 116 scenes where the first image is taken as a reference and subsequent images in a sequence are used to form pairs with increasing difficulty. This dataset can also be further separated into 57 sequences containing large changes in illumination and 59 with large changes in viewpoint.

**Repeatability.** Following [28], we compute the repeatability score for a pair of images as the number of point correspondences found between the two images divided by the minimum number of keypoint detections in the image pair. We report the average score over all image pairs.

**Matching score (M-score).** We follow the definitions given in [9, 63]. The matching score is the average ratio between ground-truth correspondences that can be recovered by the whole pipeline and the total number of estimated features within the shared viewpoint region when matching points from the first image to the second and the second image to the first one.

**Mean Matching Accuracy (MMA).** We use the same definition as in [11] where the matching accuracy is the average percentage of correct matches in an image pair considering multiple pixel error thresholds. When reporting the MMA, *i.e.* the average score for each threshold over all image pairs, we exclude as in [11] a few image sequences having an excessive resolution. Furthermore, we also report the MMA@3, *i.e.* the MMA for a specific error threshold of 3 pixels.

## 4.2 Parameter study

**Impact of $N$.** We first evaluate the impact of the patch size $N$ used in the repeatability loss $\mathcal{L}_{rep}$, see Equation 3. It essentially controls the number of keypoints as the loss ideally encourages the

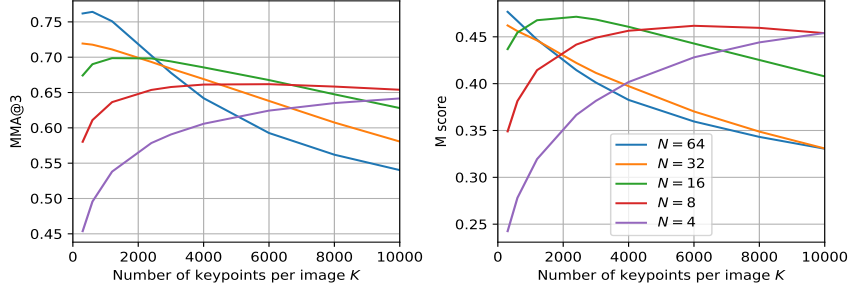

Figure 5: MMA@3 and M-score for different patch sizes $N$ on the HPatches dataset, as a function of the number of retained keypoints $K$ per image.

| Repeatability | Reliability | Keypoint score | MMA@3 | M-score |
|:---:|:---:|:---:|:---:|:---:|
| | ✓ | $\boldsymbol{R}_{ij}$ | $0.588 \pm 0.010$ | $0.361 \pm 0.011$ |
| ✓ | | $\boldsymbol{S}_{ij}$ | $0.639 \pm 0.034$ | $0.432 \pm 0.033$ |
| ✓ | ✓ | $\boldsymbol{R}_{ij}\boldsymbol{S}_{ij}$ | $\mathbf{0.688 \pm 0.009}$ | $\mathbf{0.470 \pm 0.011}$ |

Table 1: Ablative study on HPatches. We report the M-score and the MMA at a 3px error threshold for our method as well as our approach without repeatability (top row) or reliability (middle row)

network to output a single local maxima per window of size $N \times N$. Figure 4 shows different repeatability maps $\boldsymbol{S}$ obtained from the same input image with various $N$. When $N$ is large, our method outputs few highly-repeatable keypoints, and conversely for smaller values of $N$. Note that the networks even learn to populate empty regions like the sky with a grid-like pattern when $N$ is small, while it avoids them when $N$ is large. We also plot the MMA@3 and the M-score on the HPatches dataset in Figure 5 for various $N$ as a function of the number of retained keypoints $K$ per image. Models trained with large $N$ outperform those with lower $N$ when the number of retained keypoints $K$ is low, since these keypoints have a higher quality. When keeping more keypoints, poor local maxima starts to get selected for these models (*e.g.* in the sky or the river in Figure 4) and the matching performance drops. However, having numerous keypoints is important for many applications such as visual localization because it augments the chance that at least a few of them will be correctly matched despite occlusions or other noise sources. There is therefore a trade-off between the number of keypoints and the matching performance. In the following experiments, and unless stated otherwise, we use $N = 16$ and $K = 5000$.

**Impact of separate reliability and repeatability.** Our main contribution is to show that separately predicting repeatability and reliability is key to improve the final matching performance. Table 1 reports the performance aggregated over 5 independent runs when (a) removing the repeatability map, in which case keypoints are defined by maxima of the reliability map, or (b) removing the reliability map and loss, *i.e.*, only using the AP loss formulation of Equation 4. In both cases, the performance drops in terms of MMA@3 and M-score. This highlights that repeatability is not well correlated with the descriptor reliability, and shows the importance of estimating the reliability of descriptors. In the following, we select an "average" model (with 0.686 MMA@3px) for all subsequent experiments.

Figure 6 shows the repeatability and reliability heatmaps obtained for a few images. Our network trained with reliability loss is able to eliminate regions that cannot be accurately matched, such as the sky 6(a,d) or repetitive patterns artificially printed on top of the pepper photography 6(c). Note that the network has never seen the artificial patterns in 6(c) during training but is still able to reject them. More complex patterns are also discarded, such as the river in 6(a), the paved ground in 6(d), various 1-D structures in 6(a,d) or the central white building with repetitive structures in 6(a). Even though the reliability appears to be high in these regions, it is in fact slightly inferior, resulting in keypoints being scored lower which are therefore not retained in the top-$K$ final output (top row of Figure 6).

**Single-scale experiments.** To assess the importance of the multi-scale feature extraction (Section 3.3), we evaluate our model at a single-scale (full image size). We obtain $0.651$ MMA@3px compared to $0.686$ MMA@3px in the multi-scale setting.

### 4.3 Comparison with the state of the art

We now compare our approach to state-of-the-art detectors and descriptors on HPatches.

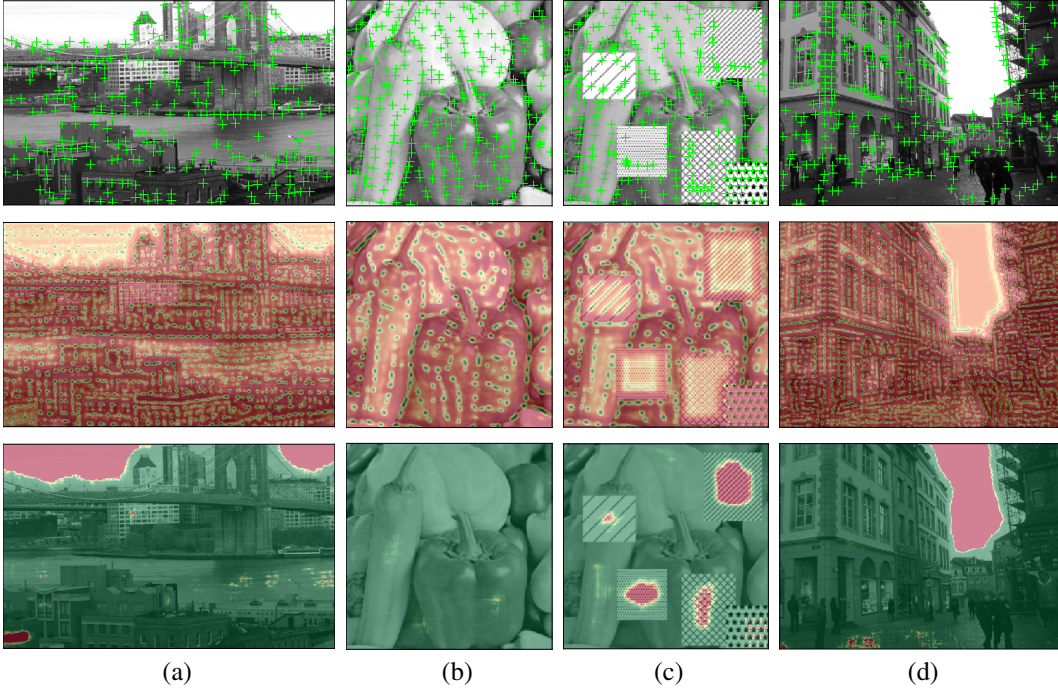

|             |             |             |             |
| :---------: | :---------: | :---------: | :---------: |
| (a)         | (b)         | (c)         | (d)         |

Figure 6: For one given input image (1st row), we show the repeatability (2nd row) and reliability heatmaps (3rd row) extracted at a single scale, and overlaid onto the original image. The reliability heatmap's color scale is enhanced for the sake of visualization. Top-scoring keypoints are shown as green crosses in the first image. They tend to avoid uniform and repetitive patterns (sky, ground, ...).

| Transformations | Data | Method | K=300 | K=600 | K=1200 | K=2400 | K=3000 |
| :--- | :--- | :--- | :--- | :--- | :--- | :--- | :--- |
| Viewpoint Perspective (VP) | graf | DoG | 0.21 | 0.0.2 | 0.18 | - | - |
| | | QuadNet | 0.17 | 0.19 | 0.21 | 0.24 | 0.25 |
| | | Ours | **0.32** | **0.38** | **0.42** | **0.45** | **0.47** |
| | wall | DoG | 0.27 | 0.28 | 0.28 | - | - |
| | | QuadNet | 0.3 | 0.35 | 0.39 | 0.44 | 0.46 |
| | | Ours | **0.62** | **0.62** | **0.65** | **0.70** | **0.71** |
| Zoom and Rotation (Z+R) | bark | DoG | 0.13 | 0.13 | - | - | - |
| | | QuadNet | 0.12 | 0.13 | 0.14 | 0.16 | 0.16 |
| | | Ours | **0.27** | **0.33** | **0.37** | **0.44** | **0.47** |
| | boat | DoG | 0.26 | 0.25 | 0.2 | - | - |
| | | QuadNet | 0.21 | 0.24 | 0.28 | 0.28 | 0.29 |
| | | Ours | **0.33** | **0.39** | **0.45** | **0.54** | **0.57** |

| Transformations | Data | Method | K=300 | K=600 | K=1200 | K=2400 | K=3000 |
| :--- | :--- | :--- | :--- | :--- | :--- | :--- | :--- |
| Luminosity (L) | leuven | DoG | 0.51 | 0.51 | 0.5 | - | - |
| | | QuadNet | **0.7** | **0.72** | **0.75** | **0.76** | **0.77** |
| | | Ours | 0.65 | 0.69 | 0.73 | **0.76** | **0.77** |
| Blur (B) | bikes | DoG | 0.41 | 0.41 | 0.39 | - | - |
| | | QuadNet | 0.53 | 0.53 | 0.49 | 0.55 | 0.57 |
| | | Ours | **0.66** | **0.67** | **0.71** | **0.75** | **0.76** |
| | trees | DoG | 0.29 | 0.3 | 0.31 | - | - |
| | | QuadNet | **0.36** | **0.39** | 0.44 | 0.49 | 0.5 |
| | | Ours | 0.28 | 0.36 | **0.45** | **0.55** | **0.6** |
| Compression (JPEG) | ubc | DoG | **0.68** | 0.6 | - | - | - |
| | | QuadNet | 0.55 | **0.62** | **0.66** | **0.67** | **0.68** |
| | | Ours | 0.40 | 0.45 | 0.54 | 0.65 | **0.68** |

Table 2: Comparison with QuadNet [49] and a handcrafted difference of gaussian (DoG) in terms of detector repeatability on the Oxford dataset, with a varying number of keypoints $K$.

**Detector repeatability.** We first evaluate our approach in terms of repeatability. Following [49], we report the repeatability on the Oxford dataset [29], a subset of HPatches, for which the transformations applied to sequences is known and include jpeg compression (JPEG), blur (Blur), zoom and rotation (Z+R), luminosity (L), and viewpoint perspective (VP). Table 2 shows a comparison with QuadNet [49] and the handcrafted Difference of Gaussians (DoG) used in SIFT [25] on this dataset when varying the number of interest points. Overall our approach significantly outperforms these two methods, in particular for a high number of interest points. This demonstrates the excellent repeatability of our detector. Note that training on the Aachen dataset may obviously helps for street views. Our approach performs well even for the cases of blur or rotation (bark, boat), while we did not train the network for such challenging cases.

**Mean Matching Accuracy.** We next compare the mean matching accuracy on HPatches with DELF [33], SuperPoint [9], LF-Net [35], mono- and multi-scale D2-Net [11], HardNet++ descriptors with HesAffNet regions [30, 31] (HAN + HN++) and a handcrafted Hessian affine detector with RootSIFT descriptor [36]. Figure 7 shows the results for illumination and viewpoint changes and the overall performance. R2D2 significantly outperforms the state of the art at nearly all error thresholds. This is at the exception of DELF in the case of illumination changes, which can be explained by their fixed grid of keypoints while this subset has no spatial changes. Interestingly, our method significantly outperforms joint detector-and-descriptors such as D2-Net [11], in particular at low level thresholds, showing that our keypoints benefit from our joint training with repeatability and reliability.

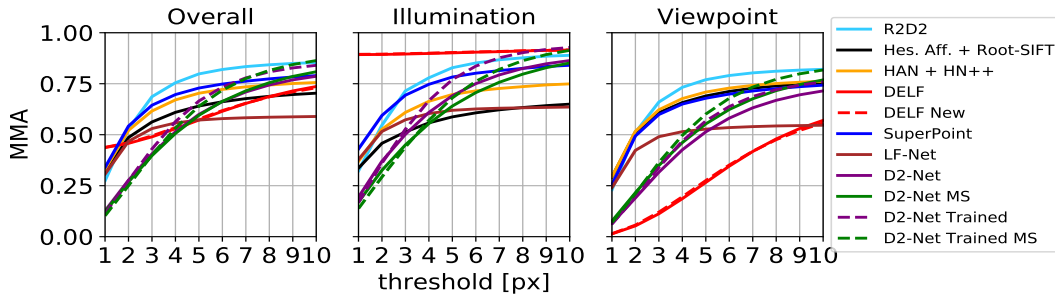

Figure 7: Comparison with the state of the art in term of MMA on the HPatches dataset.

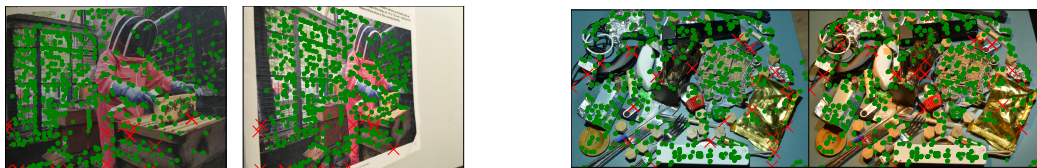

Figure 8: Sample results using reciprocal nearest matching. Correct and incorrect correspondences are shown as green dots and red crosses, respectively.

| Method | #kpts | dim | #weights | 0.5m, 2° | 1m, 5° | 5m, 10° |
|---|---|---|---|---|---|---|
| RootSIFT [25] | 11K | 128 | - | 33.7 | 52.0 | 65.3 |
| HAN+HN [31] | 11K | 128 | 2 M | 37.8 | 54.1 | 75.5 |
| SuperPoint [9] | 7K | 256 | 1.3 M | 42.8 | 57.1 | 75.5 |
| DELF (new) [33] | 11K | 1024 | 9 M | 39.8 | 61.2 | 85.7 |
| D2-Net [11] | 19K | 512 | 15 M | 44.9 | **66.3** | **88.8** |
| **R2D2**, $N = 16$ | 5K | 128 | 0.5 M | **45.9** | 65.3 | 86.7 |
| **R2D2**, $N = 8$ | 10K | 128 | 1.0 M | **45.9** | **66.3** | **88.8** |

Table 3: Comparison to the state of the art on the Aachen Day-Night dataset [45] for the visual localization task. The last row is performed with an increased number of channels.

| Training data | | | | HPatches | Aachen Day-Night | | |
|---|---|---|---|---|---|---|---|
| W | A | S | F | MMA@3px | 0.5m, 2° | 1m, 5° | 5m, 10° |
| ✓ | | | | 0.669 | 43.9 | 61.2 | 77.6 |
| ✓ | ✓ | | | 0.689 | 42.9 | 60.2 | 78.6 |
| ✓ | ✓ | ✓ | | 0.667 | 42.9 | 61.2 | 84.7 |
| ✓ | ✓ | | ✓ | 0.686 | 43.9 | 63.3 | **86.7** |
| ✓ | ✓ | ✓ | ✓ | 0.719 | **45.9** | **65.3** | **86.7** |

Table 4: Ablation study for the training data. W=web images; A=Aachen-day images; S=Aachen-day-night pairs from automatic style transfer; F=Aachen-day real images pairs. For W,A,S we use random homographies; for F optical flow.

**Matching score.** At an error threshold of 3 pixels, we obtain a M-Score of $0.453$ compared to $0.335$ for LF-Net [35] and $0.288$ for SIFT [25], demonstrating the benefit of our matching approach.

**Qualitative results.** Figure 8 shows two examples with a drastic change of viewpoint (left) and illumination (right). Our matches cover the entire image and most of them are correct (green dots).

### 4.4 Applications to visual localization

We additionnaly provide results for the visual localization task on the Aachen Day-Night dataset [45], as in D2-Net [11]. This corresponds to a realistic application scenario beyond traditional matching metrics. The goal is to find the camera poses in night images (not included in training), given the images taken during day in the same area with their known poses. We follow the "Visual Localization Benchmark" guideline: we use a pre-defined visual localization pipeline based on COLMAP [51, 52], with our matches as input. They serve to reconstruct a SfM model in which test images are registered. Reported metrics are the percentages of successfully localized images within 3 error thresholds.

For the localization task, we include an additional source of data, denoted as S, comprising night images automatically obtained from daytime Aachen images by applying style transfer. In Table 3, we compare our approach to the state of the art on the Aachen Day-Night localization task. Our approach outperforms all competing approaches at the time of submission. Table 4 shows the impact of the different sources of training data, with $N = 16$ and $K = 5000$ kpts/img (same settings as the last row but one in Table 3). We first note that training only with random web images and random homographies already yields high performance on both tasks: state-of-the-art on HPatches, and significantly better than SIFT, HAN, and SuperPoint for the localization task, showing the excellent generalization capability of our method. Adding other data sources leads to small performance gains.

We point out that our network architecture is significantly smaller than other networks (up to $15\times$ less weights) while also generating much less keypoints per image. Our keypoint descriptors are also much more compact (128-D only) compared to SuperPoint [9], DELF [33] or D2-Net [11] (resp. 256-, 1024- and 512-dimensional descriptors).

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
