[Reviews · NeurIPS 2019]

Reviewer 1



Strengths: Novel idea, promising results, overall clear written. Experiments: Authors provided detailed evaluation of the method and performed ablation study on the influence of separately repeatability and matching reliability, experiments with different transformations. Based on the novelty of the approach, performance results and evaluation, I recommend paper to be accepted.

Reviewer 2



Although the approach is somewhat incremental, I believe this is an interesting contribution for a fundamental task. The paper is well presented and experimental results are convincing. In my opinion the main technical contribution of the paper is given by the fact that the proposed network is able to estimate (at the same time) a repeatability map as well as a local descriptors associated with a discriminativeness confidence map. This leads to descriptors that can be accurately matched with high confidence. This result is obtained by relying on a metric learning procedure based on approximated average precision, which seems novel.

Reviewer 3



Novelty over [8,10] is limited. They too use a single backbone to learn a detector and descriptor which influence each other. Losses for repeatability and reliability are interesting, though. In balance, those are sufficient advances for the paper to add to our knowledge of keypoint detection and description. The "peakiness over patches" objective in Sec 3.1 is reminiscent of bucketing in SFM, where it has empirically been known to ensure a good distribution of keypoints for accurate pose estimation (for example, see "Visual odometry" by Nister, or ORB-SLAM). There might be a connection to explore here, or to state in a discussion. In Fig 2, use “confidence”, not “confidency”. Perhaps notation clarity: S’_U is the saliency map in I’ warped by the inverse of U. A dense descriptor is leaned by [6] using contrastive loss, not triplet loss as stated in Sec 3.2.

[Author Response · NeurIPS 2019]

We thank the reviewers for their valuable feedback, recognizing our work as an "*interesting contribution for a*
*fundamental task*" (R2) with "*clear contribution over prior state of the art*" (R3) that will "*definitely impact future work*"
(R1). The novelty in the losses is unanimously praised (R1, R2, R3), as well as the "*promising*" (R1) and "*convincing*
*results*" (R2, R3) that we present. We answer their main concerns below and will update the final version accordingly.

**Contributions.** We will follow R2's suggestion to improve the presentation of the major contributions w.r.t the literature.
Regarding R3's statement on limited novelty, we stress that SuperPoint [8] starts the learning of the keypoint detectors
and descriptors at different stages, while the *crux* of our approach is that we learn both of them jointly from scratch
(therefore without introducing any bias). One of our contributions is to show how this can be done efficiently and
without relying on arbitrary synthetic data and annotations as in [8].

Compared to D2-Net [10], another one of our contribution is to highlight the importance of treating repeatability and
reliability as separate entities represented by their own respective score maps. Our novel AP-based reliability loss allows
us to estimate patch reliability during training according to the AP metric while simultaneously optimizing for the
descriptor. In a single batch, each patch is typically compared to one positive versus thousands of other negative patches.
In contrast to "*Predicting matchability*" by Hartmann *et al.* (R3) that predicts reliability given fixed descriptors, our
novel loss tightly couples descriptors and reliability estimates.We will add a discussion in the related work. We believe
that this capability cannot be achieved with the standard contrastive and triplet losses used in prior work. Overall, these
advances are made possible by our novel losses that are unlike any of the ones used in [8,10,18,32,46].

**Single-scale and inference time (R1).** We have evaluated our model at a single-scale (full image size), in the same
settings as in Figure 4 ($N = 32$ and $K = 3000$ kpts/img). We obtain $0.695$ MMA@3px compared to $0.725$ MMA@3px
in the multi-scale settings. On a Tesla P100 GPU, it takes about 20 seconds to process a 1M pixel image (all scales,
with a scaling factor equal to $\sqrt[4]{2}$). Computing with a single-scale (full size) requires 30% of the total time, i.e., 6s.

**Training data and cross-dataset experiments (R1,R3).** To clarify, we use three sources of data to train our method:
(a) distractors from a retrieval dataset [35] (*i.e.* random web images), for which we build a synthetic image pair by
applying a random transformation (homography and color jittering), (b) images from the Aachen dataset [42,44] with
the same synthetic strategy to build a pair, and (c) pair of nearby views from the Aachen dataset where we obtain a
pseudo ground-truth using optical flow (Section 3.3). Note that we *do not* use any image from HPatches at training.

In order to further study performance on other datasets, R1 suggested to use AMOS Patches. However, AMOS
only evaluates for patch retrieval without the detection phase and thus not properly evaluates our approach. Instead,
we provide new results for the visual localization task on the Aachen Day-Night dataset, as in D2-Net [10]. This
corresponds to a realistic application scenario beyond traditional matching metrics. The goal is to find the camera poses
in night images (not included in training), given the images taken during day in the same area with their known poses.
We follow the "Visual Localization Benchmark" guideline: we use a pre-defined visual localization pipeline based on
COLMAP, with our matches as input. They are used to reconstruct a SfM model in which test images are registered.
Reported metrics are the percentages of successfully localized images within three error thresholds.

| Method | #kpts | dim | #weights | 0.5m, 2° | 1m, 5° | 5m, 10° |
|---|---|---|---|---|---|---|
| RootSIFT [23] | 11K | 128 | - | 33.7 | 52.0 | 65.3 |
| HAN+HN [28] | 11K | 128 | 2 M | 37.8 | 54.1 | 75.5 |
| SuperPoint [8] | 7K | 256 | 1.3 M | 42.8 | 57.1 | 75.5 |
| DELF (new) [30] | 11K | 1024 | 9 M | 39.8 | 61.2 | 85.7 |
| D2-Net [10] | 19K | 512 | 15 M | 44.9 | **66.3** | **88.8** |
| **R2D2**, $N = 16$ | 5K | 128 | 0.5 M | **45.9** | 65.3 | 86.7 |
| **R2D2**, $N = 8$ | 10K | 128 | 1.0 M | **45.9** | **66.3** | 88.8 |

Table 3. Comparison to the state of the art on the Aachen Day-Night dataset for the visual localization task. The last row is performed with an increased number of channels

| Training data | | | | HPatches | Aachen Day-Night | | |
|---|---|---|---|---|---|---|---|
| W | A | S | F | MMA@3px | 0.5m, 2° | 1m, 5° | 5m, 10° |
| ✓ | | | | 0.665 | 43.9 | 61.2 | 77.6 |
| ✓ | ✓ | | | 0.685 | 42.9 | 60.2 | 78.6 |
| ✓ | ✓ | ✓ | | - | 42.9 | 61.2 | 84.7 |
| ✓ | ✓ | | ✓ | 0.691 | 43.9 | 63.3 | **86.7** |
| ✓ | ✓ | ✓ | ✓ | - | **45.9** | **65.3** | 86.7 |

Table 4. Ablation study for the training data. W=web images; A=Aachen-day images; S=Aachen-day-night pairs from automatic style transfer; F=Aachen-day real images pairs. For W,A,S we use random homographies; for F optical flow.

For the localization task, we include an additional source of data, denoted as S, comprising night images automatically
obtained from daytime Aachen images by applying style transfer. In Table 3, we compare our approach to the state of
the art on the Aachen Day-Night localization task. Our approach outperforms all competing approaches at the time of
submission. Table 4 shows the impact of the different sources of training data, with $N = 16$ and $K = 5000$ kpts/img
(same settings as the last row but one in Table 3). We first note that training only with random web images and random
homographies already yields high performance on both tasks: state-of-the-art on HPatches, and significantly better than
SIFT, HAN, and SuperPoint for the localization task, showing the excellent generalization capability of our method.
Adding other data sources leads to small performance gains.

We point out that our network architecture is significantly smaller than other networks (up to $15\times$ less weights) while
also generating much less keypoints per image. Our keypoint descriptors are also much more compact (128-D only)
compared to SuperPoint, DELF or D2-Net (resp. 256-, 1024- and 512-dimensional descriptors).

**Code (R2).** We will release the code upon acceptance.

[Meta-Review · NeurIPS 2019]

The work tackles the problem of local feature detection and description, and proposes a novel loss function for learning both in an end-to-end pipeline. The reviewers are uniformly aligned in favor of acceptance. The (very limited) initial concerns were addressed by the rebuttal.